# Integrating beyond Content: A Framework for Infusing Elementary STEM-Focused Schools Components into Full-Service Community Schools

Erin Peters-Burton [1,*], Kathleen Provinzano [2], Kristin L. K. Koskey [2] and Toni May [2]

1   Center for Social Equity through Science Education, George Mason University, Fairfax, VA 22030, USA
2   School of Education, Drexel University, Philadelphia, PA 19104, USA; ktp37@drexel.edu (K.P.); kk3436@drexel.edu (K.L.K.K.); tas365@drexel.edu (T.M.)
*   Correspondence: epeters1@gmu.edu

**Abstract:** Learning through an integrated STEM framework has been shown to provide elementary students with numerous advantages over learning through isolated content instruction. All students, however, do not have access to high quality STEM instruction for reasons such as living in an under resourced community or their elementary school teachers feeling unprepared in STEM. For the purpose broadening STEM participation, this conceptual paper proposes a thoughtful integration of two interdisciplinary yet separate educational initiatives: STEM-focused elementary schools and full-service community schools (FSCS). In this conceptual manuscript, each educational initiative is first described independently. Then an explanation of how the central tenets of STEM-focused elementary schools and FSCS overlap is presented. Resultantly, a proposed model for integrating the two educational initiatives (FSCS^eSTEM) is depicted using a rigorous design-based research methodology. This conceptual piece ultimately demonstrates that it seems prudent to consider integrating not only content in elementary schools, but also well-researched and established educational initiatives for the possibility of expanding STEM opportunities for all. FSCS^eSTEM is one such attempt at a conceptual model proposed for future research and practice.

**Keywords:** STEM schools; integrated STEM curriculum; full service community schools; broadening STEM participation

## 1. Introduction

As societal and technological advancements quickly grow globally, being STEM literate is becoming increasingly important [1,2]. Learning through an integrated STEM framework, rather than single subject instruction, has been found to improve cognitive performance [3], provide more entry points to academic learning [4], and offer a more complex understanding of authentic phenomena [5]. However, participation in STEM fields is disproportionately low for Black, Indigenous, and People of Color (BIPOC) [6,7], suggesting that access to quality STEM instruction is not universal for all students. While there is no singular method for successfully engaging all students in STEM, the literature demonstrates that more meaningful and robust STEM opportunities can facilitate a rise in students' STEM interest and academics in early grades [8–10], particularly in BIPOC communities [11]. A systemic literature review found that elementary teachers are open to the idea of the integration of STEM content areas but do not yet feel prepared to teach STEM content [12].

Past literature has proposed that one potential way to help improve STEM literacy for all could be the integration of two distinct, but well-aligned, educational initiatives: STEM-focused schools and full-service community schools (FSCS) [13]. While this suggestion was offered in the literature, no proposed framework for how to integrate these two school structures has been presented. Thus, the purposes of this conceptual piece are to first

describe both STEM-focused schools at the elementary level and FSCS key characteristics and then provide a potential framework for how aspects of these two educational initiatives could be integrated to effectively broaden participation in STEM by building upon the strengths of each other. In doing so, we explain two conceptual frameworks focused on STEM components [14] and FSCS Pillars [15] and demonstrate how the integrated ideas could be operationalized using a design-based research approach [16].

## 2. Literature Review

### 2.1. Inclusive STEM Schools

From the beginning of his administration, former President Obama announced a concerted effort to build more STEM-focused high schools and increase the number of expert STEM teachers across the U.S. [17]. Rather than focusing on building already established selective STEM high schools, schools in Obama's STEM for All program were to be high schools accessible to all. These inclusive STEM schools had no academic admission standards and were open to all students who applied [18,19]. Inclusive STEM schools have a mission to admit a diverse student population and support all students, many of whom had little STEM experience prior to entering the school. The Authors [20] studied eight exemplar inclusive STEM high schools in different regions of the U.S. and identified 14 critical components that were common across these successful schools. This research demonstrated that although the inclusive STEM schools were located across the country and had a variety of themes and missions, they all had common components that could be transferred to other schools wanting to expand their integrated STEM offerings.

With the relative success of inclusive STEM secondary schools [20–22], this STEM educational movement began to progress to younger children in elementary schools [23,24]. Although the Authors [20] laid the foundation for critical components of successful inclusive STEM schools, many of the components could not be applied to elementary schools, due to the differences in organization, student developmental needs, and the content specialty level of teachers. Informed by the work on inclusive STEM high schools [20,25], the Authors [14] studied five exemplary STEM-focused elementary schools that had diverse student populations and found 17 critical components across school, which were grouped into five themes: (a) *school purpose and process*, (b) *community relations*, (c) *school staff*, (d) *school STEM resources*, and (e) *STEM programs*. The 17 Inclusive STEM-Focused Elementary School Critical Components, displayed as italicized phrases within text and listed in Table 1, are explained by corresponding theme below.

**Table 1.** Inclusive STEM-Focused Elementary School Critical Components Organized by Theme.

| School Purpose and Process | |
| --- | --- |
| *Inclusive STEM mission* | • Prominent theme of inclusive STEM learning organizes school design and decision making;<br>• School mission is to serve a diverse student body;<br>• Shared expectation that all students can be successful STEM learners;<br>• Some liberty within the district for a STEM focus unique from other elementary schools. |
| *Climate of intellectual safety* | • Trust, respect, and a culture of continual learning among and between students and the staff;<br>• Risk-taking and failure are presented as inherent in learning and improvement. |

**Table 1.** *Cont.*

| School Purpose and Process | |
|---|---|
| *Evidence-based improvement* | • Leadership and staff rely on evidence to refine STEM programs using design/implement/evaluate cycles;<br>• Multiple information sources inform data-driven decision making. |
| *Distributed leadership* | • Teachers have professional freedom in classrooms and decision-making input across the school;<br>• Top-down approaches are seen as incompatible with improvement-oriented, inquiry-based STEM community. |
| **Community Relationships** | |
| *Community engagement in STEM* | • Students and staff members engage community in STEM-related programs and events;<br>• Families, community members, students, and school personnel participate together in STEM activities. |
| *Supporting STEM partnerships* | • External partnerships broaden student STEM learning opportunities inside and outside the classroom;<br>• Partners provide STEM resources and expertise;<br>• Partners provide opportunities for students to engage with STEM professionals and authentic STEM experiences. |
| **School Staff** | |
| *Teachers develop and refine core curricula* | • Teachers create lessons and projects foundational to the school model and responsive to local context and student interest;<br>• Teachers use constant pilot/revise/test cycles with lessons, practices, and programs;<br>• Teachers develop ownership of and sustain STEM-focused curricula. |
| *Teachers as STEM educators* | • Teachers opt into the STEM-focused school design and mission;<br>• STEM pedagogical content knowledge is built through professional development provided by external STEM partners in the early years of a STEM school;<br>• Over time, teachers develop professional STEM identity, and ongoing professional development is self-directed and collaborative. |
| *Dedicated STEM staff* | • School receives resources or flexibility to dedicate staff to support STEM-focused offerings. |
| **School STEM Resources** | |
| *Technology used to support STEM* | • Technology is used as tool in the inquiry process;<br>• Students use technology to collect data, analyze data, share findings, and communicate ideas. |
| *School physical setting* | • School physical setting is used to immerse students in STEM-related themes and activities. |
| **STEM Program** | |
| *Interdisciplinary STEM lessons* | • Most lessons integrate STEM disciplines;<br>• Majority of lessons address multiple subject areas;<br>• School offers fully interdisciplinary STEM classes. |
| *Participation in STEM practices* | • Many lessons engage students in authentic STEM practices, typically inquiry or project-based learning;<br>• Lessons involve active and hands-on learning;<br>• Student interest in STEM is supported through participating in STEM practices. |
| *Widespread use of design cycle* | • Design cycle is used in lessons throughout the curriculum and across grades;<br>• Design cycle is used to guide and support student STEM learning. |

**Table 1.** *Cont.*

| STEM Program | |
| --- | --- |
| *21st century skills used for STEM learning* | • Collaboration, communication, creativity, and critical thinking are essential to students as STEM practitioners and as citizens;<br>• Students are given opportunities to practice and improve these skills. |
| *High-level STEM content* | • Students engage with increasingly complex STEM content;<br>• STEM content spans throughout curricula and builds across grade levels;<br>• STEM content goes beyond district and state requirements and beyond standard elementary curricula. |
| *Student ownership of learning* | • Students use a continuous improvement approach in their own learning and develop a growth mindset;<br>• Mistakes are seen as part of learning, and failure is not stigmatized;<br>• Students develop agency as learners and are encouraged to reflect on their learning process. |

### 2.1.1. School Purpose and Process

Exemplar STEM-focused elementary schools' have stakeholders who can clearly explain their purpose and the process by which they achieve their goals, categorized by four critical components. Schools have an *inclusive STEM mission* that serves to organize school design and decision-making. The school's mission is to serve a diverse student body with the expectation that all students can be successful STEM learners. To accomplish this, the school has some liberty (which varies over time) within the district to provide a STEM focus unique from other elementary schools. They have a *climate of intellectual safety* in which the school emphasizes trust, respect, and a culture of continual learning as foundational to a STEM community. Staff present risk taking and failure as inherent in learning and improvement and they promote a respectful tone and style among and between students and staff. The school conducts *evidence-based improvement.* Leadership and staff rely on evidence as they continually refine STEM programs. Data-driven decision-making, informed by multiple information sources, is carried out through a design/implement/evaluate cycle. *Distributed leadership* at these schools provides teachers with professional freedom in classrooms and decision-making input across the school. Top-down approaches are seen as incompatible with an improvement-oriented and inquiry-based STEM community.

### 2.1.2. Community Relationships

Exemplary STEM-focused elementary schools regularly reach out to the community, as evidenced by the critical component of *community engagement in STEM*. Students and school staff members engage community and family members in STEM-related activities, such as school exhibitions and projects. These schools also invite community members to become partners (*supporting STEM partnerships*). External partnerships broaden student STEM learning opportunities inside and outside the classroom. Community partners provide STEM resources, expertise, and opportunities for students to engage with STEM professionals and in authentic STEM experiences.

### 2.1.3. School Staff

At the core of an exemplary STEM-focused elementary school is its educators, who provide much of the curriculum and educate each other on STEM practices. *Teachers develop and refine core curricula* that are foundational to the school model and responsive to local context and student interest. Using a constant process of pilot/revise/test with lesson design, practices, and programs, teachers develop ownership of and work to sustain the school's STEM-focused curricula. These schools set up an environment where *all teachers are STEM educators.* Teachers opt into the STEM-focused school design and mission. During the first years of the STEM-focused elementary school, teachers participate in support provided by external STEM partnerships to develop STEM pedagogical content knowledge. Over time,

teachers develop a strong professional STEM identity and ongoing professional development becomes self-directed and collaborative. STEM-focused elementary schools also have at least one *dedicated STEM staff* member whose responsibility is to keep improving STEM opportunities for students via partnerships and teacher professional development. The school receives resources and has the flexibility to dedicate staff to support STEM-focused offerings, such as magnet coordinators or STEM lab teachers.

### 2.1.4. School STEM Resources

This theme refers to the settings and systemic resources that the schools use to promote opportunities for children to engage with STEM ideas. Students use *technology to support STEM* as part of the STEM inquiry process, including the collection of data, the analysis of data, sharing findings, and communicating ideas. The *physical school setting* is important as well because it is used to immerse students in STEM-related themes and activities while enriching the learning environment. Examples of physical resources include art installations, local wildlife settings, and school gardens.

### 2.1.5. STEM Program

STEM programming in inclusive STEM-focused elementary schools focus on building habits of mind and STEM practices. Most lessons are *interdisciplinary STEM lessons*, integrating knowledge and methods of one or more STEM disciplines, in classrooms and in STEM-focused classes or workshops. While some instruction can be discipline-specific (typically math and reading), most lessons address multiple subject areas. Many lessons offer students opportunities to *participate in authentic STEM practices*, typically through inquiry or project-based learning. Such lessons provide opportunities for active and hands-on learning. Participation in these practices helps support and sustain student interest in STEM. The schools have a *widespread use of design cycles* in many lessons, throughout the curriculum and across grades. This cycle serves as an orienting device for students and is used explicitly to guide and support student learning experiences. *Skills for the 21st century are used in STEM learning*, and practices such as collaboration, communication, creativity, and critical thinking are valued as skills essential for students' success as STEM practitioners and as STEM-fluent citizens. Thus, students are frequently given opportunities to practice and improve these skills. Students also engage with *increasingly complex STEM content* throughout the curriculum that builds across grade levels. This includes complex STEM content that goes beyond district and state requirements and beyond what is offered in standard elementary curricula. The school supports teachers in *student ownership of learning*. Students are supported in using a continuous improvement approach in their own learning, develop a growth mindset, and develop agency as learners. Staff emphasize that mistakes are part of learning and encourage students to continually reflect on their own learning process.

Although many STEM-focused elementary schools offer high quality programs to diverse student populations, there are two issues that prevent broadening STEM participation in schools as they are currently structured. First, STEM-focused elementary schools are typically the only schools of their kind in a district. They offer a lottery for admissions to keep a high level of diversity, but that means many students and parents who want their child to attend a STEM-focused elementary school do not get the opportunity. In the schools that the Author and colleagues studied, there were often three times more students applying for admission than the schools could enroll [14]. Second, these schools are schools of choice. If a parent is not informed about admission processes, the opportunity to attend a STEM-focused school is lost to the child. Admission patterns in some STEM-focused schools trend toward more white students applying over time [26]. Although the intention is for equal access, there are currently systemic barriers to STEM opportunities for all.

In an effort to increase STEM opportunities in elementary schools, we propose that alternative educational and structural approaches be considered. In addition to building STEM-focused elementary schools as single schools of choice in a school district, educational

leaders should use leverage points in neighborhood schools that already exist (i.e., FSCS) and are compatible with the critical components found in exemplary STEM-focused elementary schools. Since none of the previously studied STEM-focused elementary schools were Full Service Community Schools, this is a potential area of integration.

### 2.2. Full-Service Community Schools (FSCS)

Far too many students attending public schools in the U.S. continue to face the enduring impact of systemic racism, generational poverty, and resource constraints that severely limits, or inhibits altogether, their opportunities of receiving an educational experience that is fair and equitable. When students attend schools in resource-rich neighborhoods, they have an access advantage over those in resource-constrained regions [27], which perpetuates cycles of disadvantage and demonstrates a pressing need to rethink how schools are structured. Recently there has been renewed national interest in place-based strategies, such as FSCS, to address the decades of educational disservice [28,29]. Approximately 8000 to 10,000 schools in the U.S. now identify as community schools [30]. These schools are designed with equity in mind and are positioned as community hubs in neighborhoods with high concentrations of poverty [31]. FSCS leverage community resources to provide a holistic approach to education that is more responsive to the needs of underserved students and families [32]; are centered on partnerships between the school and local stakeholders; and provide educational, health, and social services to under-resourced communities [33].

A public investment in school improvement with the potential to mitigate the negative consequences experienced by children living in highly concentrated areas of poverty, FSCS "purposefully partner with youth organizations, health clinics, social service agencies, food banks, higher education institutions, businesses, and others to meet students' and families' academic and non-academic needs" [34]. Through these partnerships, FSCS provide wraparound services that facilitate students' opportunities to learn while also strengthening overall community well-being [35–37]. Services include enhanced educational programming, mental and behavioral health services for families, extracurricular activities, adult education opportunities, job training, and meeting basic needs (e.g., food pantries, clothing). Unlike conventional schools, FSCS are the centralized site for services, positioning them as neighborhood focal points [32] that extend the traditional relationship between school and community.

FSCS can take many forms and their contextually responsive nature make it so that no two schools look the same. Large-scale FSCS approaches include those operating in collaboration with The Children's Aid Society (National Center for Community Schools), and the University of Pennsylvania's Netter Center for Community Partnerships (University Assisted Community Schools). Expansive descriptions of these models have been detailed in the literature [31,38,39]. Many other FSCS operate with support from the state (e.g., in New Mexico—the New Mexico Community Schools Act) or school district (e.g., the Los Angeles Unified School District). Funding and implementation efforts look different from school to school; however, the FSCS strategy is centered around four pillars that are flexible and receptive to localized context and need.

#### 2.2.1. FSCS Pillars

Like the critical components found in STEM-focused schools, FSCS employ professional practices related to instruction, leadership, and service coordination that are vital to the advancement of equitable educational opportunities for underserved students. Maier et al. [15] explained that FSCS use some combination of four essential pillars: (a) *integrated student supports*, (b) *expanded learning time and opportunities*, (c) *family and community engagement*, and (d) *collaborative leadership and practices*. Table 2 operationally defines each pillar. These pillars are fundamental to the effectiveness of FSCS, as they undergird the conditions and practices necessary for providing students with high-quality learning opportunities. However, the presence of these supports alone does not make a school a FSCS. According to the Community School Playbook, "key differentiating factors include the way

in which site-based needs are identified, how the services are provided and coordinated, and their integration with other community school pillars" [40].

**Table 2.** FSCS Pillars Operationally Defined.

| Pillar | Description (Community School Playbook, 2021) |
|---|---|
| *Integrated student supports* | A dedicated staff member coordinates support programs to address out-of-school learning barriers for students and families. Mental and physical health services support student success. |
| *Expanded and enriched time and opportunities* | Enrichment activities emphasize real-world learning and community problem solving. After-school, weekend, and summer programs provide academic instruction and individualized support. |
| *Active family and community engagement* | Schools function as neighborhood hubs. There are educational opportunities for adults, and family members can share their stories and serve as equal partners in promoting student success. |
| *Collaborative leadership and practices* | Parents, students, teachers, principals, and community partners build a culture of professional learning, collective trust, and shared responsibility through site-based leadership teams and teacher learning communities. |

### 2.2.2. FSCS Research

Research on the effectiveness of FSCS is in the emergent stage but has grown significantly over the last decade [32]. Findings from a meta-analysis [41] showed FSCS programming is associated with reducing risky behavior, lowering dropout rates, and increasing the academic performance of students. These studies primarily focused on the impact of quantifiable student outcomes, such as standardized test scores, mainly in mathematics and English language arts [15,42–44] or the frequency of parental and community engagement in the school, use of services, and the correlation of those variables with student outcomes [45]. While limited evidence of FSCS impact on science achievement or other STEM-related outcomes exists, the few case studies that have been conducted show that students attending FSCS in elementary or middle school perform statistically stronger in science and STEM academic outcomes when compared to matched peers [13,42]. The scarcity of research conducted in this area further highlights the contribution of this conceptual piece to the broader literature.

### 3. Integrating Elementary STEM Themes from Critical Components with FSCS Pillars

This review of the literature demonstrates that inclusive STEM-focused elementary school initiatives and FSCS improvement strategies are becoming increasingly popular but operating parallel to one another. We posit that there are many overlapping and complementary features in both STEM schools and FSCS, and a conceptual model can clarify the intersections for a more efficient operationalized integration. Most STEM engagements with young people have generally "taken the form of various one-off experiences that in the short-term support content learning, as well as lead to affective outcomes, such as interest and inspiration" [46]. However, without intentional, long-term innovative and emerging learning environments (i.e., FSCS), these series of short interventions run counter to evidence that "deeper programmes over the course of months to years that interact not only with the young person but also with key components within their wider

learning ecology are able to measurably impact STEM aspirations" [46]. As a method of broadening STEM participation, we propose an educational model that thoughtfully integrates STEM-focused elementary school critical components with FSCS pillars. Because effective elementary STEM-focused schools and FSCS share similar features and each have demonstrated student educational successes, we argue that an integrated educational model would allow for each approach to build on the strengths of the other. We call this model FSCSeSTEM.

Figure 1 illustrates the intersection of the inclusive STEM-focused elementary school themes and FSCS pillars. In an integrative educational model, the four FSCS pillars would serve as an ideal foundation for expanding elementary STEM-focused schools into new, more inclusive environments. Elementary STEM-focused school critical components are shown building on each aligned FSCS pillar to clearly demonstrate the already overlapping components.

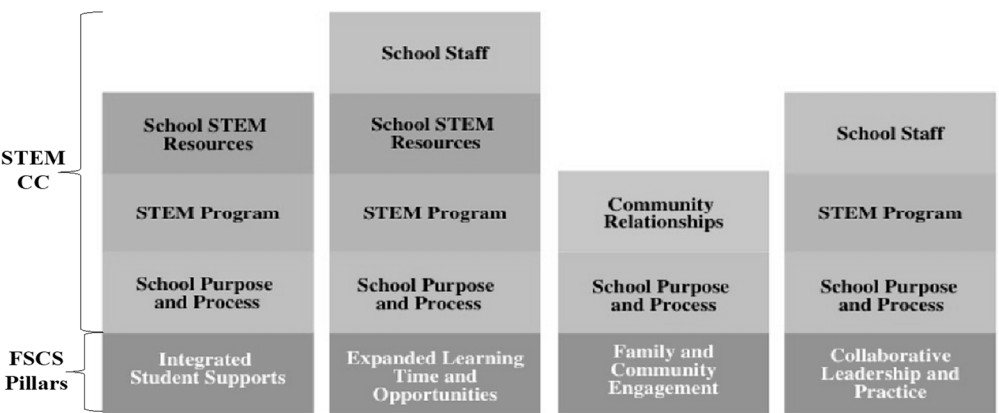

**Figure 1.** Overlap Between Elementary STEM-Focused School Themes from Critical Components and FSCS Pillars.

By carefully integrating the five themes of STEM-focused schools into an educational model with FSCS foundational pillars, the potential for sustained, unparalleled opportunities for inquiry-based STEM learning that connects in- and out-of-school education within the context of local community exists. Thus, it stands to reason that our proposed FSCS STEM-focused (FSCSeSTEM) integrative educational model is both timely and important. If the individual approaches are taken in tandem, they have the potential to offer a learning ecosystem that can positively affect the STEM education landscape overall and particularly for the nation's most vulnerable youth. It is, therefore, important to consider how an integrated FSCSeSTEM elementary school could come to fruition and what an effective working model might look like in practice.

### 3.1. Integrated Student Supports

Exemplary STEM-focused elementary schools intentionally integrate student supports with an understanding that students, including BIPOC students, may not have had any exposure to STEM experiences. For example, STEM-focused elementary schools promote a climate of intellectual safety for all and refine supports for students through evidence-based improvement cycles. For example, one of the elementary schools studied had prominently displayed bulletin boards populated with failed student engineering design projects that included explanations of how the failure led to improved learning in the student's own words [14]. This communicated a clear message to all students that everyone has failures and they are opportunities to learn rather than mistakes. Through intentional design, *integrated student supports* provide routinely marginalized students attending under-resourced schools with access to opportunities that are often only available to their wealthier peers [47]. In FSCS, these services "provide a tool for building constructive relationships" that are designed to "address gaps in care for youth in need of support" [15]. Support

comes in many forms but is situated in shared principles of care that are collaborative, culturally competent, and strengths/asset-driven [15,48].

### 3.2. Expanded Learning Time and Opportunities

Elementary STEM schools create expanded learning time and opportunities via the themes of *school purpose and process*, *STEM programs*, *school STEM resources*, and *school staff*. For example, elementary STEM schools expand STEM experiences for students by having teachers who are the designated STEM leaders, and they are responsible for teaching a "special" class in addition to the traditional art and physical education classes. Grade level teachers at traditional schools may drop off their students for art class. In elementary STEM schools, the grade level teacher team teaches with the STEM specialist teacher. In doing so, the grade level teacher receives in vivo professional development on STEM instruction that they take back to the grade level classroom. This creates expanded STEM opportunities that are integrated into the grade level classroom instruction or into after school clubs. Within the context of FSCS, the concept of expanded learning time and opportunities is extended to include vantages for students to interact with community partners primarily through out-of-school, informal learning activities [15]. These opportunities are designed to provide students attending schools in high poverty neighborhoods with activities that complement those occurring within the classroom and allow students to pursue their own interests. This is important because research has shown students attending under-resourced schools do not have access to such enrichment activities compared to their peers' attending schools in affluent neighborhoods [49]. Thus, an important connection between expanded learning opportunities and FSCS has been established [50].

### 3.3. Family and Community Engagement

Elementary STEM schools rely on the skills and expertise of the greater community for their success. They bring STEM-relevant community members into the schools to share expertise with teachers for the purposes of instruction and assessment. The schools also reach out to STEM-related organizations for teacher professional development. Teachers shadow STEM professionals and create lessons with authentic STEM learning to teach the following school year. Maier and colleagues (2017) studied family and community engagement in community schools within the context of three strands: (a) parent support of student learning, (b) family and community participation in school, and (c) family and community engagement. Overall, they found "community schools are particularly well positioned to have strong family and community engagement programs that can be bolstered by their collaborative practices, expanded learning opportunities, and integrated student services" [15], though this is an area of research that is still emerging in the field of community schools. FSCS strategies do, however, make explicit that to significantly improve outcomes for students, collaborations with families and community partners are essential [51]. Without such a focus on out-of-school factors (i.e., nutrition, housing, healthcare, safety), underserved students in resource constrained neighborhoods remain largely ignored and disproportionally impacted [52].

### 3.4. Collaborative Leadership and Practice

Elementary STEM schools employ distributed leadership, and the administration gives teachers professional freedom and decision-making input for school-wide initiatives. School-wide changes occur in the elementary STEM schools from grass-root efforts that parallel an engineering design cycle. Ideas for new initiatives are developed from data collected at the classroom level and designed for one or two classrooms. They are implemented with an intention to collect evidence that will assess the initiative. Once the evidence is evaluated, successful initiatives are deployed broadly across the school. Evidence is collected for the first few cycles to inform the sustainability of the initiatives. FSCS adhere to principals of distributive leadership in a similar way. Collaborative leadership practices in FSCS are described as a "mediating factor", which is necessary for the

aforementioned three pillars to work [15]. The overlapping partnerships inherent in FSCS contexts requires principals to forgo top-down approaches in lieu of leadership practices that are inclusive of a wide-ranging stakeholder voice that extends beyond professionals in the school [53]. Resultantly, FSCS readily position collaborative leadership practices as a mechanism for building trust amongst a multitude of partners committed to improving student learning and well-being.

## 4. FSCS and STEM Integration (FSCS^eSTEM) Using Design-Based Research

Integration of the Inclusive Elementary STEM-Focused Critical Components involves "creating conditions that support educators in making innovations into working infrastructures for organizing learning activities" [54]. The existing interconnected pieces in the FSCS need to be considered in this process to "help us identify potential leverage points for change" [54]. Each FSCS is uniquely designed as responsive to local needs. As such, integration must be place-based and achieved at the school community level. Place-based education is a process that meaningfully connects students with their local environment to address real-time community issues while also focusing on broader educational requirements [55]. Collaborating to shape the integration as place-based so that it is not "*done to*" but rather "*done along with*" the FSCS will result in a more sustainable working infrastructure [54]. Effective practices for when partnering with practitioners in change efforts are documented in the literature [54,56].

Design-based research (DBR) approaches tend to reflect these evidence-based practices and iteratively support multiple objectives, varying contexts, and multi-layered interactions. This research framework draws from methods commonly applied in engineering, product design, user-centered design, and action research [57,58]. It is described as an iterative cycle of designing, testing, evaluating, and reflecting [59]. The researcher *and* practitioner are engaged throughout the cycle taking place in a real-world context [60]. DBR thus is more iterative, participatory, and contextualized as compared to traditional experimental designs where the researcher drives the study in a controlled environment [60].

Multiple DBR approaches exist in the literature. For the purpose of our illustration, we apply Bannan-Ritland's [16] Integrative Learning Design Framework (ILDF) used in learning environments. ILDF consists of four phases: Phase 1—Informed Exploration (*IE*) of theory, the literature, and needs analysis to inform the design; Phase II—Enactment (*E*) of the design in practice; Phase III—Local Impact (*LI*) to formatively test to refine the design based on systematic evaluation of results; and Phase IV—Broader Impact (*BI*) to disseminate results to expand "diffusion, adoption, and adaptation" and seek continued input from local and broader stakeholder groups [16]. This iterative design approach has been used in similar contexts working with practitioners to support constructing, implementing, and systematically studying instructional initiatives in learning environments nested in complex systems [16,61–63].

Collaborating with practitioners (intended users) is a key component to ILDF to support their goals and increase adoption and use of the innovation in practice [16]. Researchers and FSCS partners collaborate from development through dissemination by engaging in the iterative four phases of the ILDF. Thus, an important component we propose in the development and implementation of an FSCS^eSTEM is identifying a dedicated liaison who will have a centralized role in launching, supporting, and sustaining the integration, much like the specialized STEM personnel critical component from the Authors' study [14]. In addition to providing program coordination, the liaison facilitates interactions among FSCS stakeholders, researchers, community-based organizations, families, students, and other relevant groups.

*4.1. Phase I—Informed Exploration*

An aim of the *IE* phase will be consensus on the STEM-focused school critical components to infuse into an FSCS. To achieve this outcome during exploration, researchers and FSCS partners collaborate and share expertise. Researchers communicate basic understandings of the critical components to schools based on the research literature through a multiple-day professional development.

Following professional development, with the support of researchers, FSCS engage in a needs assessment using the *eSTEM School Components Rating Inventory* [64]. FSCS partners and researchers identify stakeholders at each school, such as community level leaders (district, STEM advocates, lead school partners), administrators, teachers, other education professionals, support professionals, parents, and students. It is recommended that at least two representatives from each stakeholder group complete the *eSTEM School Components Rating Inventory* [64]. The inventory consists of six critical components broken down into three to five indicators that each stakeholder rates from strongly agree to strongly disagree. For example, one indicator of the School Purpose and Process component is *climate of intellectual safety*. Table 3 displays the instructions and questions related to this indicator. Once all components are rated, stakeholders typically notice differences in rating the same indicator, which leads into the sharing of information about school processes.

**Table 3.** Example Item on the eSTEM School Components Rating Inventory.

| School Purpose and Process | | | | | |
|---|---|---|---|---|---|
| **The following statements refer to Climate of Intellectual Safety.** *Please consider the following definition when rating the statements below.* The school has a prominent theme of inclusive STEM learning that serves to organize the school design and decision making. The school's mission is to serve a diverse student body with the expectation that all students can be successful STEM learners. To accomplish this, the school has some liberty (which varies over time) within the district to provide a STEM focus unique from other elementary schools. | | | | | |
| **This school has the following components . . .** | **Strongly Disagree (1)** | **Disagree (2)** | **Don't Know (3)** | **Agree (4)** | **Strongly Agree (5)** |
| a. Trust, respect, and a culture of continual learning among and between students and the staff. | | | | | |
| b. Risk taking and failure are presented as inherent in learning and improvement. | | | | | |
| Comments regarding **Climate of Intellectual Safety** component: | | | | | |

When agreement of the rating for each indicator among stakeholders has been established, school personnel can determine which component indicators are already present at the school, and which component indicators can be prioritized for improvement for 3-year, 5-year, and 10-year plans. Findings are used to then purposefully select between two to three Exemplar STEM-Focused Elementary Schools who were involved in the critical component study to engage with through site visits as further exploration in Phase I (see Table 4 for a list of schools). Needs assessment findings inform which Exemplary STEM-focused elementary school sites would be most aligned to the key areas of change identified.

**Table 4.** Exemplary STEM-Focused Elementary Schools.

| School | Year Established | Location | Grade | Example Distinction of Exemplary |
|---|---|---|---|---|
| Brentwood Magnet Elementary School of Engineering | 1957 (became STEM in 2008) | Raleigh NC | PK-5 | North Carolina STEM Model School of Distinction (2016) |
| Douglas L. Jamerson, Jr. Elementary School | 2008 | St. Petersburg FL | K-5 | Magnet Schools of America Award of Merit (2007) |
| Summit Road STEM Elementary | 2011 | Reynoldsburg OH | K-4 | National Blue Ribbon School Distinction (2016) |
| Walter Bracken STEAM Academy | 1961 (became STEAM in 2009) | Las Vegas NV | PK-5 | Five-star rating on the Nevada School Performance Framework (since 2010) |
| Weaver Lake Elementary | 1991 (became STEM in 2004) | Maple Grove MN | PK-5 | FETC STEM Excellence Award (2015) |

It is recommended that multiple FSCS partners from across schools (e.g., teachers, FSCS liaison, Principals, District Science Coordinators, District Level Executive Directors) and the researchers engage in a multi-day site visit at each exemplary school selected. During the site visits, FSCS partners and researchers orient themselves with a whole-school tour and can break into groups to investigate ways in which the exemplar STEM-focused elementary schools carry out the critical components of interest. By gaining an overview of the school's activities, then delving into each component more deeply, FSCS partners can take into account whole school design and consider implications for change on the whole community. In addition to observing classes and extracurricular activities, small groups of FSCS partners and researchers interview administrators, teachers and conduct focus groups with students, families, and community partners. Findings from the *eSTEM School Components Rating Inventory* [64] and site visit can be used to (a) refine 3-year, 5-year, and 10-year plans and (b) inform the design of more STEM opportunities at each school.

*4.2. Phase II—Enactment*

An aim of the *E* phase is collaborative development of logic models outlining the detailed design for infusing the critical components identified in Phase I and intended STEM outcomes to iteratively testing out the critical components in practice. Logic models have been used in community initiatives as a communication tool to facilitate reaching consensus and providing a blueprint for monitoring activities and outcomes [65,66]. Prior NSF DRK-12 funded STEM-focused elementary (#blinded) and high school (#blinded) studies have developed logic models for infusing the critical components school-wide. Schools are complex systems also requiring integration of non-linear systems approaches to map inter-relationships at a deeper level among components of STEM-focused FSCS, in this case [67,68].

Six questions outlined by Mertens and Wilson [66] for guiding logic model building can be tailored to facilitate this process:

1. What conditions in the FSCS have resulted in the need for a STEM-focused FSCS?
2. What are the FSCS needs and assets?
3. What are the specific desired results (outputs, outcomes, and long-term impacts)?
4. What are the factors that facilitate or impede change?
5. What strategies are needed to achieve the intended results?
6. Why do the FSCS partners believe these strategies will lead to these intended results?

A backward design approach is effective in guiding stakeholders through these questions starting with intended results to first identify outcomes and then activities leading to those outcomes following the strategy to "Plan backward and implement forward" [69]. Once the logic models are developed and adopted, short, iterative tests of STEM oppor-

tunities can be performed to refine foundational components, participants and resources, STEM program outcomes, and outputs using evidence-based processes.

### 4.3. Phase III—Local Impact

Aims of the *LI* phase will be a further refined design for each school based on systematic study of the effectiveness of the integration of the critical components into the curriculum on STEM outcomes [61,70]. The purpose of the *LI* phase is to implement the individualized plans of the FSCS partners and collaboratively examine the practicality, useability, and fidelity of the plan. Given this feedback, the FSCS will begin to scale-up the parts of the plan that worked as intended and revise the parts of the plan that need improvement. For example, in the elementary STEM school study [53], teachers and administrators noted that students were not checking books out of the library and were concerned that it could be contributing to low reading scores. In the *IE* phase, school leaders conducted a needs analysis and found that one of the reasons was that the library was not in the daily pathways of teachers and students. In the *E* phase, the teachers and administration formed a new idea, moving selected book series into teachers' room for all students to check out. This allowed for a more convenient location, but also helped students meet teachers in the school that they may not have known. In the *LI* phase, a few teachers tested this idea by taking a few book series into their classrooms and formed a checkout system. The school collected data from teachers through interviews and from students through questionnaires about the books that they read to measure success of the small test. These data informed what worked and what did not work in the Local Impact phase. When the test teachers were confident that the change would work for the whole school, they moved to the *BI* phase.

### 4.4. Phase IV—Broad Impact

In the *BI* phase, the process and results of the project are disseminated. Although this is presented as Phase IV, communication of the integration process and results is ongoing across phases so that findings can be used to further refine the design. As described by Bannan-Ritland (2003), "the adoption (and adaptation) of the researched practices and interventions" and "consequences of use" [16] are important to communicate across stakeholder groups, as is seeking continuous input through interactive communications. The use of multiple mediums for dissemination is important to reach a broader audience and might include but are not limited to:

- Dialoguing with community partners
- Video showcases
- In-person showcases
- Photo voices
- Email communications
- Board or advisory meetings
- Newsletters

- Social media postings
- Blogs
- Broadcast media
- Communications to share through the FSCS network
- Professional development offerings
- Conference presentations
- Peer-reviewed publications

In this phase, the new FSCS$^{eSTEM}$ not only enacts change efforts school-wide and disseminates the process and results, but the school itself transforms into a demonstration school community for others to explore. Dissemination and dialoguing provide opportunities for local stakeholders to gain an understanding of the integration and potentially further their participation. It also provides possibilities for FSCS Network and STEM stakeholders at the broader level to evaluate potential transfer ideas and lessons learned in other FSCS contexts. Figure 2 graphically depicts the DBR cycle in this FSCS$^{eSTEM}$ context.

**Phase I**

**Informed Exploration of eSTEM Critical Components**

**PRODUCT:** *Data and community informed multi-year action plans.*

o  Engage in professional development on eSTEM Critical Components
o  Conduct a needs assessment using *eSTEM School Components Rating Inventory*
o  Consensus on which key eSTEM Critical Components to integrate
o  Visit Exemplary STEM-focused Elementary Schools
o  Develop 3-year, 5-year, and 10-year plans for integrating eSTEM Critical Components
o  Design eSTEM opportunities

**Phase IV**

**Evaluation of Broader Impact of Integration**

**PPRODUCT:** *Dissemination of model and results to broader FSCS and STEM learning communities.*

o  Develop a dissemination plan to reach the FSCS and STEM learning communities through multiple mediums
o  Share your FSCS^eSTEM model and evaluation results with FSCS and STEM learning communities
o  Showcase the school and lessons learned for others interested in transferring the model

**Phase II**

**Enactment of eSTEM Critical Components**

**PRODUCT:** *Refined logic models based on iterative testing of key eSTEM Critical Components in practice.*

o  Construct logic models outlining the design for Critical Components integration
o  Conduct short, iterative tests of eSTEM opportunities using evidenced-based improvement processes
o  Refine logic models based on findings

**Phase III**

**Evaluation of Local Impact of Integration**

**PRODUCT:** *Evaluation impact study results to further refine integration of eSTEM Critical Components into practice.*

o  Conduct outcomes study to determine impact of integrating the Critical Components
o  Collaborative interpret the findings
o  Develop action plans based on findings
o  Share the model, evaluation results, and lessons learned with the local community through multiple mediums

**Figure 2.** Integrating STEM-Focused Elementary Critical Components into an FSCS Using Integrative Learning Design Framework.

## 5. Conclusions

As discussed throughout this conceptual paper, public education in the U.S. runs rife with reform initiatives, and many well-intentioned improvement strategies result in failure [71]. Data collected over the last decade have indicated that often, large-scale school reform initiatives designed to address educational inequities and positively impact student achievement have had little desired effect [72]. STEM-related programming is no different. K-12 STEM education continues to experience an array of challenges in the U.S., especially for BIPOC students attending schools in high-poverty neighborhoods [73]. Increasing the quality of STEM education has long been a national priority but because no "predetermined structural pathway" [74] exists for access to STEM learning opportunities in places where resources are limited, school districts are largely left to figure out their own STEM approaches. This is problematic because common amongst the range of STEM-school archetypes are magnet-style schools or charter-based schools with selective admission processes that tend to exclude marginalized students [74]. Further, these schools usually operate at the middle or high school levels, even though research tells us STEM interest usually develops during the elementary school years [75]. To nurture and sustain young children's' curiosity with STEM, and to broaden underrepresented groups' access to quality STEM learning opportunities, the authors have advanced a framework merging the critical components of inclusive STEM elementary schools within the context of a broader whole-school reform strategy (FSCS), requiring extensive collaboration across the school-community landscape.

Drawing on Bannan-Ritland's (2003) ILDF, the proposed FSCS^eSTEM encourages researchers and FSCS partners to collaborate from development through dissemination by engaging in the iterative ILDF phases. As a result, FSCS^eSTEM provides a pathway for conceptualizing sustainable school reform within the spirit of collaboration and couched

within a message of decolonization. Schools are often thought to be "key social institutions that can support social and economic mobility" [76], yet structural inequalities and patterns of resegregation, especially for Black and Latino students and students living in poverty, persist [77]. For these reasons, school and neighborhood stakeholders have turned to FSCS as a strategy to reduce inequality by way of a whole-child, contextually responsive approach. FSCS "hold great promise for addressing racialized and classed inequalities by divesting in deficit-oriented and assimilationist schooling to instead advocate for developmentally appropriate, asset-based, and race and class conscious educational practices" [76]. Incorporating the critical components of inclusive STEM elementary schools within the confines of neighborhood FSCS pushes the boundaries of equitable school reform by assembling two justice-centered models typically operating in isolation. Ideally, the result is a framework that intentionally corrects the miseducation of racially and/or economically minoritized youth in schools.

From a practice-based perspective, FSCS^eSTEM demands the work be in collaboration *with* school and community stakeholders to create systems-level change that pervades every aspect of schooling. As demonstrated in Figure 1, FSCS^eSTEM draws its energy from disrupting singularly focused, one-off reform initiatives that tend to perpetuate systemic inequities instead of addressing them. The recursive nature of the work requires intensive collaboration across several different stakeholder groups so that STEM learning experiences can be integrated into the school curriculum and actualized in elementary classrooms. Potential implementation challenges associated with FSCS^eSTEM are to be expected and should be considered at the forefront. For example, although teachers express a desire to integrate STEM into their teaching, they often feel underprepared to do so because of their limited content knowledge in these areas [78]. Typically trained as generalists who instruct all subject areas, elementary teachers tend to lack confidence in teaching STEM-related subject matter [79], which resultantly leads to less time spent on science and mathematics instruction [78]. FSCS^eSTEM accounts for this reality by meeting teachers where they are in terms of STEM readiness and providing on-going professional development and support as they learn to integrate the elementary STEM critical components into their teaching.

To realize equity in STEM education necessitates continued research and practice. The literature has documented STEM schools that enable BIPOC students to be successful in STEM [18] and STEM schools that have persistent barriers for BIPOC students [74]. It is still unclear what factors and contexts can help STEM schools support and provide access at a young age for all students. Attempts to combine interdisciplinary educational frameworks, particularly those with a focus on underserved students, have the potential to inform environments in unique ways to offer STEM opportunities for all. FSCS^eSTEM is one such attempt at a conceptual model proposed for future research and practice.

**Author Contributions:** Conceptualization, E.P.-B., K.P., K.L.K.K. and T.M.; writing—original draft preparation, E.P.-B., K.P., K.L.K.K. and T.M.; writing—review and editing, E.P.-B., K.P., K.L.K.K. and T.M. All authors have read and agreed to the published version of the manuscript.

**Funding:** This research received no external funding.

**Conflicts of Interest:** The authors declare no conflict of interest.

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
