# Peer review of "Integrating beyond Content: A Framework for Infusing Elementary STEM-Focused Schools Components into Full-Service Community Schools"

_education, doi:10.3390/educsci12080511_

Round 1
Reviewer 1 Report
I enjoyed reviewing this manuscript on the STEM integration (infusion) framework. Overall, I feel this is a well-written article with many valid suggestions.
The authors begin the paper by clearly describing Inclusive STEM Schools. The critical components in Table 1 help to define these schools' common characteristics. FSCS are also clearly defined for your audience. Were any of the Inclusive STEM schools studied previously considered FSCS? Demographically, do they share any similar characteristics?
To increase equity and diversity in STEM, I'm surprised to see culturally responsive STEM pedagogies and a multicultural STEM curriculum not listed specifically as integrated student supports. One could argue that these are just as significant as (and necessary for) utilizing place-based educational methods.
Finally, I appreciate the limitations and additional justifications provided in the Conclusions section. This model would require sincere buy-in and collaboration from many different stakeholders to see long term success. However, this suggestion for an infused model is overdue and needed.
Author Response
None of the Inclusive STEM schools studied previously were FSCS. We have added a sentence on page 6 to reflect this. “Since none of the previously studied STEM-focused elementary schools were Full Service Community Schools, this is a potential area of integration.”
We added the phrase “that had diverse student populations” on page 2 to reflect their demographic similarities.
In the prior STEM-focused elementary studies we did not find multicultural STEM curriculum to be a component from our empirical work. This is perhaps because the focus of the research was on STEM practices, and they were inherently culturally responsive because the school recruitment was deliberately diverse.
Thank you for your helpful comments.
Reviewer 2 Report
Thank you for inviting me to be a reviewer of the manuscript entitled Integrating Beyond Content: A Framework for Infusing Elementary STEM-Focused Schools Components into Full-Service Community Schools. This document is really impressive in terms of your efforts to demonstrate the power of your study.
The current study examines the teaching of the STEM framework in schools and the incorporation of concepts into education. The authors describe STEM and FCSC conceptual frameworks and their operationalization based on a research-based approach.
The introductory chapters of this study present a theoretical basis based on a Literary review, which is focused on the concept of STEM in schools. The authors also address the FSCS concept and its pillars. They also focus on the integration of STEM and FSCS using design-based research. And they describe the individual phases in detail: phase 1: Informed Exploration, phase 2: Enactment, phase 3: Local Impact, phase 4: Broad Impact. The relationships of the individual hphases are clearly shown in Figure 3. The conclusion of the entire study follows. At this point, I would expect more emphasis on the discussion of the findings of the presented study. I would further describe and specify the following objectives of the study.
In the study, I see great potential for further follow-up research.
However, some passages of the study are very descriptive and lengthy. This is sometimes confusing. Therefore, I would suggest shortening and simplifying them.
This study refers to 70 scientific references, resources and publications. The first references used are self-citations. The references used are current and of sufficient quality, and are a suitable tereotic basis for this study.
This study represents a contribution in this area of research.
The basic ideas of the submitted manuscript are fascinating and interesting.
Author Response
Thank you for your valuable comments. The purpose of this article was to propose conceptual frameworks that haven’t yet been studied that could constitute a new area of research. Therefore, we did not yet do an empirical study from which to discuss findings.
We have gone through the manuscript and shortened and clarified where possible, while still responding to the other reviewers.
Reviewer 3 Report
Years ago having written a grant somewhat related to the topic of the manuscript it is obvious the authors have done their homework. They have presented a well-thought out framework for full-service community schools. Their tables and figures enhance the manuscript because they succinctly illustrate the various aspects of STEM-focused elementary schools, the overlap of those elementary schools with full-service community schools, and ultimately Figure 3 where an integrative learning design framework is used to show the integration of STEM-focused elementary schools and FSCS.
They have covered all aspects of what it takes for FSCS to be successful with supportive literature and incorporation of design-based research, specifically an integrative learning design framework, logic design, and all the way to the broad impact with numerous suggested dissemination possibilities
There are a number of places where English style and design need to be addressed:
p1. no cap on Former in reference to President Obama
Throughout the manuscript when they refer to the different aspects of the integrative learning design framework that is used to show the integration of STEM-focused elementary schools and FSCS caps are used for the different aspects. These should not be capitalized, they are not proper nouns. Italicizing all of them would make them stand out more.
When Authors is used at the beginning of a sentences, "the" should precede "authors".
p. 2 The word "the" is missing 7 lines down on the first full paragraph - and the content... and the next to the last line of the same paragraph - ...within the text...
p. 3 2nd full paragraph, approximate middle of the paragraph - ...teachers participate in support provided... - no s on support
p. 4-6 Spacing problems on the first column of Table 1
p. 5 Teachers STEM Educators - third line, ...partners in the early years... - add, the
p. 6 - 8th line - Should House, be Houser?
p. 6 Remove the "s" on trends on line 13
p. 7 - first full paragraph, line 1 - ...with the potential to ... - add "the"
p. 8 - last paragraph on the page - first line - ...review of the literature... - add "the"
p. 10 - 2nd full paragraph, line 7 - ...drop off their students for art class. In elementary...
p.11 - first line - ...schools rely on...
p. 12 - line 7 - ...as place-based so that it is not...
p. 14 - spacing in Table 3
Author Response
Thank you for your helpful suggestions and taking the time to give us specific direction. We have edited the manuscript accordingly to respond to all of your suggestions.